# Cellular and Molecular Network Characteristics of *TARM1*-Related Genes in *Mycobacterium tuberculosis* Infections

**DOI:** 10.3390/ijms251810100

**Published:** 2024-09-20

**Authors:** Li Peng, Hanxin Wu, Liangyu Zhu, Jieqin Song, Weijiang Ma, Lei Zhong, Weijie Ma, Rui Yang, Xun Huang, Bingxue Li, Suyi Luo, Fukai Bao, Aihua Liu

**Affiliations:** 1Yunnan Province Key Laboratory of Children’s Major Diseases Research, School of Basic Medical Sciences, Kunming Medical University, Kunming 650500, China; pengli@kmmu.edu.cn (L.P.); wuhanxinn@163.com (H.W.); zhuliangyu1216@163.com (L.Z.); 20211707@kmmu.edu.cn (J.S.); 20221745@kmmu.edu.cn (W.M.); zhonglei202202@163.com (L.Z.); 13385177456@163.com (W.M.); 20230029@kmmu.edu.cn (R.Y.); 18309907586@163.com (X.H.); libingxue1@kmmu.edu.cn (B.L.); 13987133389@163.com (S.L.); 2Yunnan Provincial Key Laboratory of Public Health and Biosafety, School of Public Health, Kunming Medical University, Kunming 650500, China

**Keywords:** *TARM1*, *Mycobacterium tuberculosis*, bioinformatics, tuberculosis, biomarker

## Abstract

Tuberculosis (TB) is a global infectious threat, and the emergence of multidrug-resistant TB has become a major challenge in eradicating the disease that requires the discovery of new treatment strategies. This study aimed to elucidate the immune infiltration and molecular regulatory network of T cell-interacting activating receptors on myeloid cell 1 (*TARM1*)-related genes based on a bioinformatics analysis. The GSE114911 dataset was obtained from the Gene Expression Omnibus (GEO) and screened to identify 17 *TARM1*-related differentially expressed genes (TRDEGs). Genes interacting with the TRDEGs were analyzed using a Gene Ontology (GO) and Kyoto Encyclopedia of Genes and Genomes (KEGG) enrichment analysis. A gene set enrichment analysis (GSEA) was used to identify the biological pathways significantly associated with a *Mycobacterium tuberculosis* (*Mtb*) infection. The key genes were obtained based on Cytoscape’s cytoHubba plug-in. Furthermore, protein–protein interaction (PPI) networks were analyzed through STRING, while mRNA–RNA-binding protein (RBP) and mRNA–transcription factor (TF) interaction networks were developed utilizing the StarBase v3.0 and ChIPBase databases. In addition, the diagnostic significance of key genes was evaluated via receiver operating characteristic (ROC) curves, and the immune infiltration was analyzed using an ssGSEA and MCPCounter. The key genes identified in the GSE114911 dataset were confirmed in an independent GSE139825 dataset. A total of seventeen TRDEGs and eight key genes were obtained in a differential expression analysis using the cytoHubba plug-in. Through the GO and KEGG analysis, it was found that these were involved in the NF-κB, PI3K/Akt, MAPK, and other pathways related to inflammation and energy metabolism. Furthermore, the ssGSEA and MCPCounter analysis revealed a significant rise in activated T cells and T helper cells within the *Mtb* infection group, which were markedly associated with these key genes. This implies their potential significance in the anti-*Mtb* response. In summary, our results show that TRDEGs are linked to inflammation, energy metabolism, and immune cells, offering fresh insights into the mechanisms underlying TB pathogenesis and supporting further investigation into the possible molecular roles of *TARM1* in TB, as well as assisting in the identification of prospective diagnostic biomarkers.

## 1. Introduction

Tuberculosis (TB) is an infectious disease that poses a significant threat to human life and health, and is primarily caused by *Mycobacterium tuberculosis* (*Mtb*). Its persistence and transmission make it one of the most severe public health challenges faced globally [1,2]. Although the number of TB cases worldwide is decreasing year by year with improvements in medical conditions and sanitation, the population of patients with TB and the disease burden due to TB are still quite large. The Global Tuberculosis Report in 2023 showed that nearly 10 million people were infected by *Mtb* in 2022, and about 1.6 million people died from the disease [3]. Owing to the complex pathogenesis of TB, the low early detection rate, and the emergence of drug-resistant bacterial strains, the disease burden is still difficult to reduce [4,5]. Therefore, it is important to identify new strategies for treating *Mtb* infections, which requires the continuous exploration of the pathogenesis and therapeutic targets.

The T cell-interacting activating receptor on myeloid cells 1 (*TARM1*), a promising new gene discovered in 2015 [6], encodes a receptor protein rich in arginine residues and is mainly expressed in monocytes and neutrophils [6]. The simultaneous activation of *TARM1* and Toll-like receptor 4 (*TLR4*) increased the release of the pro-inflammatory cytokines TNF-α and IL-6 from macrophages and neutrophils [6]. With respect to TB treatment research, only one study reported that *TARM1* plays a significant role in macrophage activation and the Th1 response, but the specific mechanism was unclear [7]. Our research group focused on exploring the anti-TB mechanism of *TARM1*. Consistent with other reports [7], our research revealed that the levels of *TARM1* expression were increased in TB patients compared to healthy individuals. In this study, we extensively analyzed the molecular network characteristics and regulation mode of *TARM1*-related genes (TRGs), which provided valuable ideas and an important basis for the study of the role of *TARM1* in TB.

In recent times, various bioinformatics techniques have gained popularity for examining microarray and high-throughput datasets, particularly for identifying differentially expressed genes (DEGs) [8,9]. Additionally, these techniques are beneficial for exploring the underlying pathways associated with numerous human disorders. Given that metabolic irregularities and the infiltration of immune cells play significant roles in the etiology of TB, this research leveraged a thorough genomic evaluation reliant on the Gene Expression Omnibus (GEO) database. The objective was to uncover the characteristics related to the immune infiltration and molecular networks of TRGs, with the aim of discovering novel early diagnostic biomarkers, while also offering new insights into the potential molecular mechanisms of *TARM1* in TB.

## 2. Results

### 2.1. Dataset Correction

This research primarily employed bioinformatics techniques to investigate the biological characteristics of *TARM1*-related DEGs (TRDEGs) in *Mtb* infections, with the overall analysis flow chart illustrated in Figure 1. Within datasets GSE114911 and GSE139825, the samples were classified into *Mtb*-infected samples (infected group) and non-infected samples (uninfected samples). The GSE114911 (Figure 2A,B) and GSE139825 (Figure 2C,D) datasets were standardized, with annotations applied to probes and other data-cleaning processes, followed by the creation of box plots to illustrate the data distribution prior to and following standardization. After standardization, the expression levels of different samples in the datasets tended to be consistent.

### 2.2. TARM1-Related Differentially Expressed Genes Associated with Mtb

The data derived from the GSE114911 dataset were categorized into infected and uninfected groups, respectively. To evaluate the gene expression in the GSE114911 dataset, the differences between the infected and uninfected groups were analyzed. The limma R package was utilized for a differential analysis of the DEGs, resulting in two distinct sets of data. There were 1201 DEGs that satisfied the criteria of |log2FC| > 0 and *p* < 0.05. A total of 594 genes were found to be up-regulated (log2FC > 0 and adjusted *p* < 0.05), while 607 genes were identified as down-regulated (log2FC < 0 and adjusted *p* < 0.05). The outcomes of this dataset’s differential analysis are illustrated in the volcano plot presented in Figure 3A.

To obtain all the TRDEGs, the intersection of all the DEGs and TRGs was considered and a Venn diagram was generated (Figure 3B). A total of 17 TRDEGs were obtained (Appendix A). The analysis of the TRDEG expression discrepancies among various sample groups (infected/uninfected) in the GSE114911 dataset was conducted, and a difference sequencing diagram (Figure 3C) was created by utilizing the R package pheatmap.

### 2.3. Gene Ontology (GO) and Kyoto Encyclopedia of Genes and Genomes (KEGG) Enrichment Analysis

GO and KEGG enrichment analyses were conducted to delve deeper into the connections among the biological processes (BPs), cell components (CCs), molecular functions (MFs), and KEGG associated with the 17 TRDEGs and *Mtb*. The findings are presented in Appendix A. The results of the GO analysis revealed that the TRDEGs were predominantly enriched in the following BP categories: pathways mediated by cytokines, cellular responses to lipopolysaccharides, molecules of bacterial origin, and biotic stimuli. Additionally, the analysis demonstrated that the TRDEGs exhibited significant enrichment in the CC categories, including the external side of the plasma membrane, tertiary granules, specific granules, and tetraspanin-rich microdomains. In terms of MFs, the TRDEGs showed enrichment primarily in cytokine activity, binding to cytokine receptors, and receptor–ligand interactions. Furthermore, the KEGG pathway analysis results revealed that 17 TRDEGs showed significant enrichment in various pathways, including the interaction between cytokines and their receptors, viral protein interactions with cytokines and receptors, the JAK-STAT signaling pathway, and the chemokine signaling pathway, as well as in diseases such as pertussis, leishmaniasis, and rheumatoid arthritis.

The findings from the GO and KEGG enrichment analyses are illustrated using bubble plots (see Figure 4A,B). Concurrently, we generated network diagrams for the BPs, CCs, MFs, and KEGG (Figure 4C,D) based on the results of the GO and KEGG analyses. Each line describes the corresponding molecule and items, with larger nodes indicating a higher number of constituent molecules. Furthermore, the combined log2FC from the GO and KEGG enrichment evaluations is presented in Figure 4E,F.

### 2.4. Gene Set Enrichment Analysis (GSEA)

To investigate the mechanisms through which TRDEGs contribute to the anti-TB mechanism, we investigated the association between the expression levels of all genes within the GSE114911 dataset, the BPs involved, the CCs affected, and the MFs performed by GSEA (Figure 5A) (Appendix A). The dataset revealed significant enrichment of all the genes in pathways such as MAPK (Figure 5B), Hedgehog (Figure 5C), JAK-STAT (Figure 5D), PI3K/Akt (Figure 5E), and NF-κB (Figure 5F). 

### 2.5. Protein–Protein Interaction (PPI), mRNA–RNA-Binding Protein (RBP), and mRNA-Transcription Factor (TF) Interaction Network Construction

We utilized the STRING database to conduct a PPI analysis on 17 TRDEGs, and established a PPI network with a minimum required coefficient exceeding 0.150 while excluding non-correlated genes. The interaction relationships were visualized, and the PPI network diagram was created using the Cytoscape software (version: 3.10.0) (Figure 6A).

Subsequently, the cytoHubba plugin was utilized to evaluate and identify eight genes (*CXCL1*, *FCGR3A*, *CXCL5*, *CCL4*, *IL2RA*, *IL36G*, *CRLF2*, and *TNIP1*) as key genes shared by the top ten genes revealed by each of the five different algorithms: MCC, MNC, EPC, Degree, and DMNC (Figure 6B,C). Next, we used the STRING database to conduct a PPI analysis for the eight identified key genes (with a minimum required interaction score of >0.150), and constructed a PPI network (Figure 6C).

Initially, we utilized the mRNA-RBP data obtained from the ENCORI database to identify RBP interactions involving these eight key genes. The regulatory network of mRNA-RBP was constructed and illustrated using Cytoscape (Figure 6D), comprising 5 key genes (*CRLF2*, *CXCL1*, *CXCL5*, *IL2RA*, and *TNIP1*); 57 RBP molecules; and 62 mRNA-RBP interaction relationships (Appendix A). Next, we retrieved the TFs that had binding affinities to the aforementioned eight key genes by consulting the ChIPBase database. The mRNA-TF regulatory network was also created and represented by Cytoscape (Figure 6E). There were 5 key genes (*CXCL1*, *CXCL5*, *FCGR3A*, *IL2RA*, and *TNIP1*) and 31 TFs, culminating in 52 interaction relationships between mRNA and TFs (Appendix A). 

### 2.6. Analysis of Differential Gene Expression and Receiver Operating Characteristic (ROC) Curves for Key Genes in the Dataset Comparing Infected and Uninfected Groups

We used a violin plot (Figure 7A) to show the differential expression levels of the eight key genes (*CXCL1*, *FCGR3A*, *CXCL5*, *CCL4*, *IL2RA*, *IL36G*, *CRLF2*, and *TNIP1*) across various groups (uninfected/infected) in the GSE114911 dataset. Our findings indicated that the expression levels of four key genes (*CCL4*, *IL2RA*, *IL36G*, and *TNIP1*) had statistically significant differences (*p* < 0.001); those of two key genes (*CXCL1* and *CXCL5*) exhibited high statistical significance (*p* < 0.01); and that of *CRLF2* was also statistically significant (*p* < 0.05). Meanwhile, we used the violin plot (Appendix A) to show the differential expression levels of eight key genes (*CXCL1*, *FCGR3A*, *CXCL5*, *CCL4*, *IL2RA*, *IL36G*, *CRLF2*, and *TNIP1*) between different groups (uninfected/infected) in the GSE139825 dataset.

Then, we plotted ROC curves to evaluate the potential of the eight key genes as biomarkers to confirm their diagnostic efficacy in TB. In the GSE114911 dataset, the area-under-the-curve (AUC) values for CCL4 (AUC = 0.876), IL2RA (AUC = 0.85), TNIP1 (AUC = 0.81), IL36G (AUC = 0.799), CXCL5 (AUC = 0.762), CXCL1 (AUC = 0.746), CRLF2 (AUC = 0.713), and FCG3A (AUC = 0.63) were assessed (Figure 7B–I), along with the performance in the GSE139825 dataset (Figure 8A–G) of CRLF2 (AUC = 0.829), CXCL1 (AUC = 0.8), CCL4 (AUC = 0.8), TNIP1 (AUC = 0.8), IL2RA (AUC = 0.743), CXCL5 (AUC = 0.743), and FCG3A (AUC = 0.629). These results indicated that the key genes *CCL4*, *IL2RA*, *TNIP1*, *IL36G*, *CXCL5*, *CXCL1*, and *CRLF2* are potential biomarkers of TB.

### 2.7. Immune Infiltration Analysis (ssGSEA and MCPCounter)

We used the ssGSEA algorithm to assess the correlation among the expression profiles of 28 immune cell types across different subgroups (infected/uninfected) in the GSE114911 dataset. Based on the results from the immune infiltration analysis, a box plot illustrating the infiltration abundance of these 28 immune cell types is presented (Figure 9A). The findings indicated that the expression levels of five kinds of immune cell types were statistically significant (*p* < 0.05). These immune cell types included activated CD4 T cells, type 1 T helper cells (Th1), effector memory CD8 T cells, type 17 T helper cells (Th17), and type 2 T helper cells (Th2). In addition, we used a heat map (Figure 9B) to show the correlation between eight key genes (*CXCL1*, *FCGR3A*, *CXCL5*, *CCL4*, *IL2RA*, *IL36G*, *CRLF2*, and *TNIP1*) and the infiltration abundance of five immune cell types (*p* < 0.05). The findings indicated that the key gene *IL2RA* had a positive correlation with immune Th17 cells, and the key gene *CCL4* was positively correlated with activated CD4 T cells. However, the key gene *FCGR3A* was negatively correlated with Th2 cells. 

The MCPCounter algorithm was utilized to assess the relationship between these key genes and the infiltrating abundance of immune cells in different subgroups (infected/uninfected) in the GSE114911 dataset (Figure 9C). Our findings indicated a correlation between the eight key genes and ten kinds of immune cells (including B lineage cells, T cells, CD8 T cells, cytotoxic lymphocytes, endothelial cells, neutrophils, fibroblasts, monocytic lineage cells, myeloid dendritic cells, and NK cells). Among them, the key genes *FCGR3A* and *CRLF2* were positively associated with neutrophils and CD8 T cells. In contrast, *CXCL5* and *TNIP1* showed a negative correlation with fibroblasts and myeloid dendritic cells.

## 3. Discussion

The main treatment for TB is antibiotic therapy. Isoniazid (INH) and rifampicin are currently the two most effective antibiotics against *Mtb* [10,11]. Still, the rise in multidrug-resistant TB (MDR-TB) that targets these antibiotics presents a significant obstacle to effective treatment and the elimination of the disease [5]. As a result, it is very important to investigate new strategies to treat TB and explore its therapeutic targets.

In this study, the molecular network characteristics and regulatory patterns of TRDEGs during *Mtb* infections were systematically analyzed using bioinformatics methods and previous studies. Through a differential expression analysis, we identified 17 TRDEGs in the GSE114911 dataset. By conducting a GO/KEGG enrichment analysis, we observed that these TRDEGs primarily focused on inflammation-related cytokine activity, cytokine-related signaling pathways, and JAK-STAT signaling pathways. In addition, a GSEA was performed, and according to the normalized enrichment score (NES), the five signaling pathways with the highest correlation were identified as JAK-STAT, NF-κB, Hedgehog, PI3K/Akt, and MAPK. The above information suggests that inflammation and cytokine responses dominate after an *Mtb* infection. 

Cytokines are small, soluble proteins synthesized by cells that influence the physiological activities of other cells, mainly in a paracrine manner. They have the ability to bind to receptors located on cell membrane surfaces, subsequently activating the JAK-STAT signaling pathway, which is crucial for cell proliferation, differentiation, and immune regulation [12,13]. In addition, NF-κB is vital in mediating cellular inflammatory and immune responses [14]. Interestingly, TRDEGs showed significant enrichment in the PI3K/Akt and MAPK signaling pathways after an *Mtb* infection. These two signaling pathways are essential for energy metabolism and survival signaling in cells, and they also intersect with glycolysis [15,16]. In recent times, researchers have increasingly recognized the important relationship between basic metabolism and the functionality of immune cells.

Immune metabolism, which pertains to metabolic reprogramming in immune cells, emphasizes the study of cellular functions critical to immune response and alterations in host cell metabolism during *Mtb* infections. These alterations play a role in regulating the expression of different cytokines and chemokines linked to the processes of inflammation resolution and suppression and the progression of TB [17,18]. After an *Mtb* infection, the aerobic glycolysis of macrophages increases (i.e., the reprogramming of glucose metabolism) and autophagy is activated by increasing lactic acid secretions, which play an anti-*Mtb* role [19]. This process depends on the activation of the AKT/mTOR signal, which can activate the NF-κB signaling pathway to inhibit *Mtb* [20]. In summary, promoting *Mtb* clearance by targeting host cell metabolism can be an effective adjunct to antibiotic therapy and can shorten its duration. This study provides support for anti-TB research targeting host metabolism.

Bioinformatics serves as a robust instrument for the identification of molecular indicators [21]. We used the “cytoHubba plugin” to screen out eight key genes (*CXCL1*, *FCGR3A*, *CXCL5*, *CCL4*, *IL2RA*, *IL36G*, *CRLF2*, and *TNIP1*), and then explored the differences in the expression and diagnostic significance of these genes in both datasets. The results indicated that, except for *FCGR3A*, the expression levels of the other seven key genes exhibited significant variation between the *Mtb*-infected and uninfected groups. The AUC in the ROC curve of these seven key genes surpassed 0.7, suggesting their potential role as biomarkers for a TB diagnosis. 

*TNIP1* is crucial for diagnosing TB in pediatric populations [22]. *CXCL1* serves as a new diagnostic biomarker for human TB [23], with a high sensitivity and specificity for both active tuberculosis (ATB) and latent tuberculosis (LTB). Both *CXCL1* [24] and *CXCL5* are important chemokines. During an *Mtb* infection, *CXCL5* is vital for attracting polymorphonuclear leukocytes (PMNs) to the infected lungs, and TB is significantly improved in the absence of *CXCL5* [25]. *CCL4* is a macrophage-derived inflammatory chemokine that can induce the chemotactic mobilization of immune cells [26]. For TB patients, after INH treatment, *CCL4* decreases and *IL2RA* increases. Therefore, changes in the expression profiles of *IL2RA* and *CCL4* throughout INH therapy represent valuable indicators for tracking the treatment of LTB [27]. *IL36G* has resulted in important progress in the study of MDR-TB [28]. In addition, through the analysis of the PPI, mRNA-RBP, and mRNA-TF interaction networks associated with these key genes, we found some RBPs and TFs that can bind to key genes, which can provide useful guidance for further investigations into the functional mechanisms in the development of TB.

To enhance our understanding of the infiltration dynamics of immune cells during *Mtb* infections and the molecular regulatory network of TRDEGs, we used an ssGSEA and MCPCounter to perform an immune infiltration analysis of these eight key genes, suggesting that the enhanced infiltration of activated B cells, Th17 cells, T cells, and Th1 cells may be associated with the development of TB. Our future research will target cellular immunity and the anti-TB mechanism of *TARM1*.

In the current study, we discuss eight key genes linked to *TARM1* involved in the onset and advancement of TB, suggesting that these genes could act as promising biomarkers and therapeutic targets. However, the limitations of this study should be considered. First, the key genes identified require further validation using clinical samples. Furthermore, the mechanisms through which these key genes operate remain ambiguous. Additional evidence is necessary to clarify the biological basis.

## 4. Materials and Methods

### 4.1. Data Download

The GSE114911 [29] and GSE139825 [30] datasets were download from the GEO database [31] (https://www.ncbi.nlm.nih.gov/geo/ (accessed on 25 March 2024)) using the R package GEOquery (version: 2.72.0) [32]. Among them, the GSE114911 dataset contained 33 *Mtb*-infected samples (infected group) and 19 non-infected samples (uninfected group). The GPL6480 Agilent-014850 Whole Human Genome Microarray 4x44K G4112F (Agilent Technologies, Palo Alto, Santa Clara, CA, USA) microarray platform was used. Dataset GSE139825 contained seven *Mtb*-infected samples (infected group) and five non-infected samples (uninfected group). The GPL10558 Illumina Human HT-12 V4.0 (Illumina Inc., San Diego, CA, USA) expression beadchip was used for all the datasets. All the samples in the datasets were from *Homo sapiens*, and these samples were included in the analysis (details in Table 1). The GSE114911 dataset was the test set and the GSE139825 dataset was the validation set.

STRING [29] is a database for searching the interactions between known and predicted proteins. The STRING database (https://cn.string-db.org/ (accessed on 25 March 2024)) was used to collect TRGs. In the STRING database, we used “TARM1” as the search keyword and set the minimum interaction score to >0.150; subsequently, a total of 151 TRGs were obtained (Appendix A).

### 4.2. TARM1-Related Differentially Expressed Genes (TRDEGs) Associated with Mtb

According to the grouping of the samples in dataset GSE114911, the samples were divided into an infected group and an uninfected group. The R package limma (version: 3.60.4) was used to analyze the differences between the genes in the different groups. The threshold values of |log2FC| > 0 and *p* < 0.05 were set as the DEGs. Genes with log2FC > 0 and adjusted *p* < 0.05 were up-regulated DEGs. Genes with log2FC<0 and adjusted *p* < 0.05 were down-regulated DEGs.

To obtain TRDEGs associated with *Mtb*, all the DEGs with |log2FC| > 0 and *p* < 0.05 obtained by a differential analysis in the GSE114911 dataset were intercrossed with TRGs, and a Venn diagram was generated to obtain the TRDEGs. The results of the differential analysis were plotted using the R package ggplot2 (version: 3.5.1) to draw volcano plots and differential ranking maps.

### 4.3. Functional Enrichment Analysis (GO) and Pathway Enrichment (KEGG) Analysis

A GO analysis [30] is a common method for large-scale functional enrichment studies, including biological processes (BPs), cell components (CCs), and molecular functions (MFs). KEGG (https://www.genome.jp/kegg/ (accessed on 25 March 2024)) [31] is a widely used database that stores information on genomes, biological pathways, diseases, and drugs. The R package clusterProfiler (version: 4.12.6) [32] was used to perform the GO and KEGG enrichment analysis of TRDEGs. The item-screening criteria of *p* < 0.05 and an FDR value (q-value) <0.05 were considered to indicate statistically significant differences.

### 4.4. Gene Set Enrichment Analysis (GSEA)

A GSEA [33] is used to evaluate the distribution trend of genes in a predefined gene set in a gene table ranked by correlation with phenotype, in order to determine their contribution to phenotype. In this study, the genes of the GSE114911 dataset were first ranked according to their log2FC value, and then the R package clusterProfiler was used to perform a GSEA on all the genes in the GSE114911 dataset. The parameters used in the GSEA were as follows: the seed was 2020, the number of computations was 1000, the minimum number of genes contained in each gene set was 10, and the maximum number of genes contained in each gene set was 500. Through the Molecular Signatures Database (MSigDB) [34], access to c2 gene sets was obtained. Cp. All. V2022.1. Hs. Symbols. The GMT [all canonical pathways] (3050) was used for the GSEA. The screening criteria of the GSEA were *p* < 0.05 and an FDR value (q-value) <0.25. 

### 4.5. Construction of Protein–Protein Interaction (PPI), mRNA-RBP, and mRNA-TF Interaction Networks

The PPI network is composed of individual proteins interacting with each other. In this study, we used the STRING database with the biological species set to human, and the confidence level was set to >0.150. After excluding unrelated genes, we constructed a PPI network and used Cytoscape to visualize it.

The following five algorithms were used to select the top 10 common TRDEGs as key genes: degree correlation (Degree), maximum neighborhood component (MNC), maximal clique centrality (MCC), density of maximum neighborhood component (DMNC), and edge percolated component (EPC) in the cytoHubba [35] plug-in.

RNA-binding proteins (RBPs) [36] play a key role in the regulation of life activities such as RNA synthesis, alternative splicing, modification, transport, and translation. Based on the StarBase v3.0 database [37], the target RBP of key genes (mRNA) was predicted, and the mRNA-RBP regulatory network was visualized by the Cytoscape software (version: 3.10.0).

Transcription factors (TFs) control gene expression by interacting with key genes (mRNA) at the post-transcriptional stage. TFs were retrieved from the ChIPBase database [38] to analyze the regulatory effects of the TFs on key genes (mRNA), and the mRNA-TF regulatory network was visualized by the Cytoscape software (version: 3.10.0).

### 4.6. Differential Expression Analysis of Key Genes

To identify the potential mechanism of action and related biological characteristics and pathways of DEGs in *Mtb* infections, we used the Mann–Whitney U test to identify the DEGs based on the GSE114911 dataset. The expression differences between the *Mtb*-infected and uninfected groups in the GSE139825 dataset were analyzed, and the results of the difference analysis were shown using the R package ggplot2 (version: 3.5.1) to draw group comparison maps.

Next, we screened the key genes based on the expression of different analysis results, and further analyzed the receiver operating characteristic (ROC) curve of the key genes in the GSE114911 and GSE139825 datasets. A ROC curve [39] is a coordinate schema-based analysis tool that can be used to select the best model, discard the second-best model, or set the best threshold in the same model. The ROC curve is a comprehensive indicator of continuous variables reflecting sensitivity and specificity, which reflects the relationship between sensitivity and specificity by the composition method. The area under the curve (AUC) is generally between 0.5 and 1. The closer the AUC is to 1, the better the diagnostic effect. When the AUC was between 0.5 and 0.7, the accuracy was low; when the AUC was between 0.7 and 0.9, the accuracy was moderate; and when the AUC was >0.9, the accuracy was high. We used R’s proc package (version: 1.18.5) to plot the ROC curves of the screened key genes in the GSE114911 and GSE139825 datasets, and calculated the AUC. The AUC was calculated to evaluate the diagnostic effect of TRDEG expression on the survival of patients with *Mtb* infections.

### 4.7. Immune Infiltration Analysis (ssGSEA and MCPcounter)

A single-sample gene set enrichment analysis (ssGSEA) [40] quantifies the relative abundance of each immune cell infiltrate. First, each infiltrating immune cell type was labeled, such as activated CD8 T cells, activated dendritic cells (DCs), natural killer (NK) cells, and a variety of human immune cell subtypes, such as regulatory T cells. Second, the enrichment scores calculated by the ssGSEA were used to represent the relative abundance of each immune cell infiltration in each sample, and the samples with *p* < 0.05 were filtered out to obtain the immune cell infiltration matrix. The difference in the immune cell infiltration abundance between the Mtb-infected and uninfected groups was shown by a group comparison plot. The R package pheatmap (version: 1.0.12) was used to draw correlation heat maps to show the correlation between immune cells and TRDEGs in Mtb and immune cells.

The infiltration of immune cells and other stromal cells of the tissue was estimated from the expression data using MCPCounter (version: 1.2.0) [41]. MCPcounter provides abundance estimates for eight immune cell populations, namely CD4^+^ T cells, CD8^+^ T cells, NK cells, B lymphocytes, monocytes, DC cells, neutrophils, and cytotoxic lymphocytes. The abundance of two non-immune stromal cell populations, namely fibroblasts and endothelial cells, was also calculated by this method. Spearman’s correlation analysis was performed based on the MCPCounter abundance estimates, as well as the expression of antigen genes, and a *p* < 0.05 was considered to indicate statistically significant differences.

### 4.8. Statistical Analysis

All the data processing and analyses in this article were based on R software (version 4.2.2). If not otherwise specified, the statistical significance of normally distributed variables was estimated by Student’s *t*-test of two independent groups for continuous variables between two groups. The Mann–Whitney U test was used to analyze the differences between variables that were not normally distributed. The Kruskal–Wallis test was used for comparisons of three or more groups. Spearman’s correlation analysis was used to calculate the correlation coefficient between different molecules. All the statistical *p*-values were two-sided if not specified, and *p* < 0.05 was considered to indicate statistically significant differences.

## 5. Conclusions

Our research integrated two GEO datasets of *Mtb* infection and systematically analyzed the immune infiltration and molecular network characteristics of TRDEGs using suitable bioinformatics tools. Through this comprehensive analysis, we identified eight potentially key genes (*CXCL1*, *FCGR3A*, *CXCL5*, *CCL4*, *IL2RA*, *IL36G*, *CRLF2*, and *TNIP1*) and their related signaling pathways. These findings contribute to a deeper understanding of the molecular mechanisms underlying *Mtb* infections and may provide valuable insights for future exploration of the anti-TB mechanism of *TARM1*. Additionally, the identified key genes and pathways may facilitate the discovery of novel diagnostic biomarkers and potential therapeutic targets for TB.

## Figures and Tables

**Figure 1 ijms-25-10100-f001:**
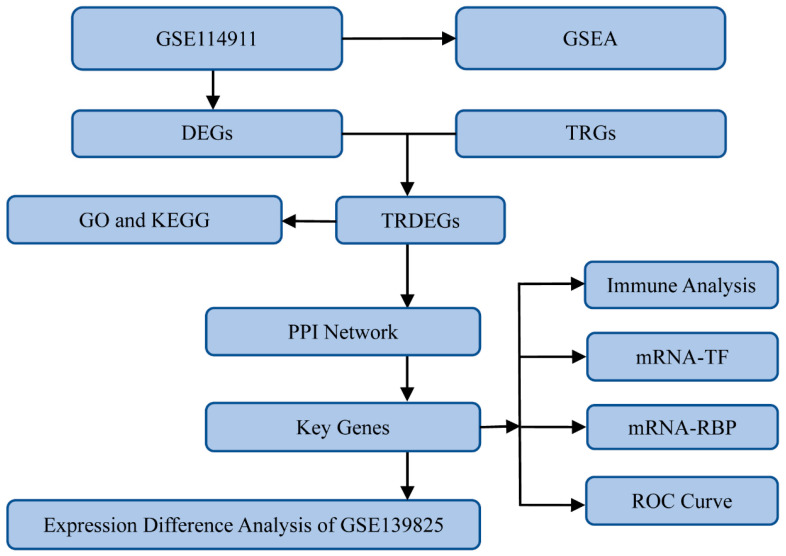
Flow chart of the study. GSEA: gene set enrichment analysis. DEGs: differentially expressed genes. TRGs: *TARM1*-related genes. TRDEGs: *TARM1*-related differentially expressed genes. GO: Gene Ontology. KEGG: Kyoto Encyclopedia of Genes and Genomes. PPI Network: protein–protein interaction network. ROC: receiver operating characteristic. RBP: RNA-binding protein. TF: transcription factor.

**Figure 2 ijms-25-10100-f002:**
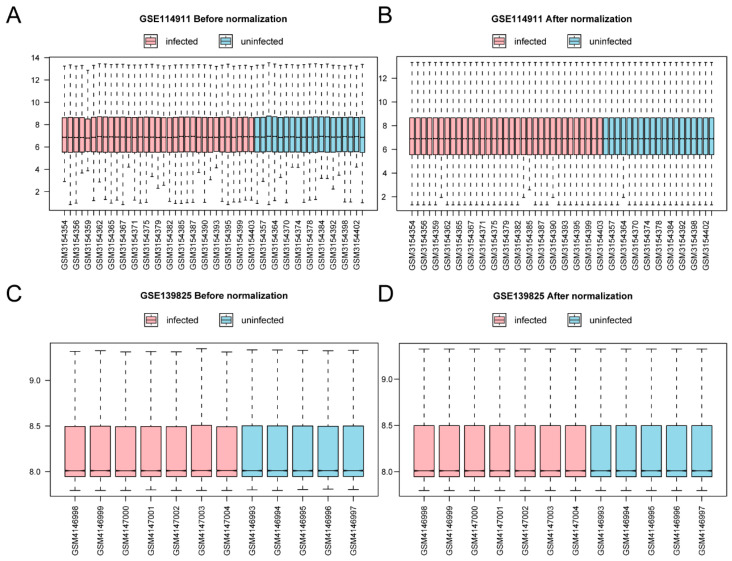
Datasets for batch processing. (**A**) Box plot of gene expression distribution between GSE114911 samples before correction. (**B**) Box plot of distribution of gene expression among corrected GSE114911 samples. (**C**) Box plot of gene expression distribution among GSE139825 samples before correction. (**D**) Box plot of gene expression distribution among corrected GSE139825 samples. *Mtb*-infected samples (infected) in red and uninfected samples (uninfected) in blue in the datasets.

**Figure 3 ijms-25-10100-f003:**
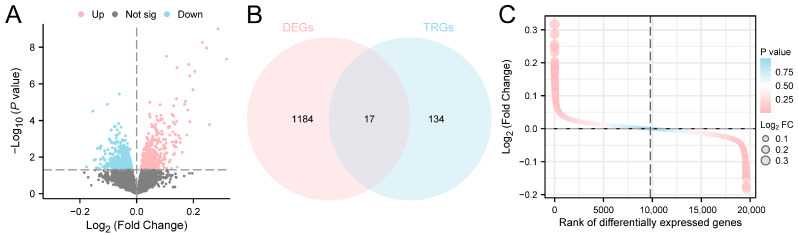
The differential gene expression analysis. (**A**) A volcano plot of the DEG analysis between infected and uninfected groups in the GSE114911 dataset. (**B**) A Venn diagram of the DEGs and TRGs in the GSE114911 dataset. (**C**) A difference sequence diagram of the TRDEGs in the GSE114911 dataset. The size of the bubbles indicates the number of genes, while the color of the bubbles reflects the size of the *p*-value. A redder hue corresponds to a smaller *p*-value, whereas a bluer hue indicates a larger *p*-value.

**Figure 4 ijms-25-10100-f004:**
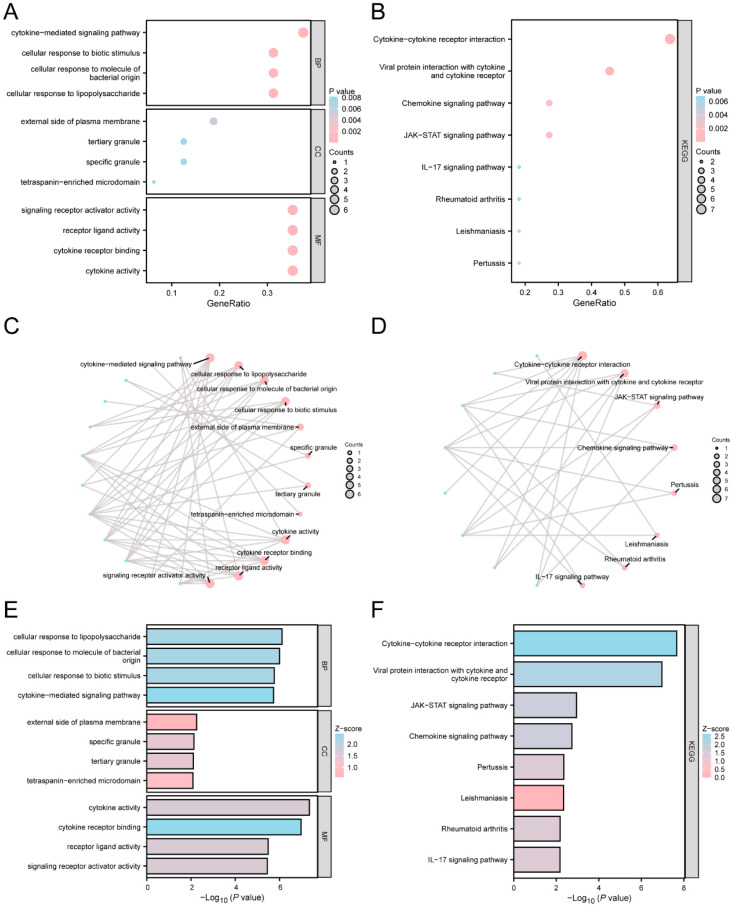
The enrichment analysis of GO and KEGG for TRDEGs. (**A**) A bubble plot of the GO enrichment analysis results for TRDEGs: BPs, MFs, and CCs. (**B**) The bubble plot represents the KEGG enrichment analysis findings for TRDEGs. The bubble size correlates with the number of genes, while the color indicates the magnitude of the adjusted *p*-value—darker red signifies smaller adjusted *p*-values, whereas darker blue indicates larger adjusted *p*-values. (**C**,**D**) The network diagram shows the findings of the GO and KEGG enrichment analyses for TRDGEs. The red nodes represent items and the blue nodes represent molecules, with attachment on behalf of the entry and molecular relationship. (**E**,**F**) Bar graphs of the TRDEGs’ GO and KEGG enrichment analysis findings. The GO and KEGG selection criteria were a *p*-value < 0.05 and an FDR value (q-value) < 0.05.

**Figure 5 ijms-25-10100-f005:**
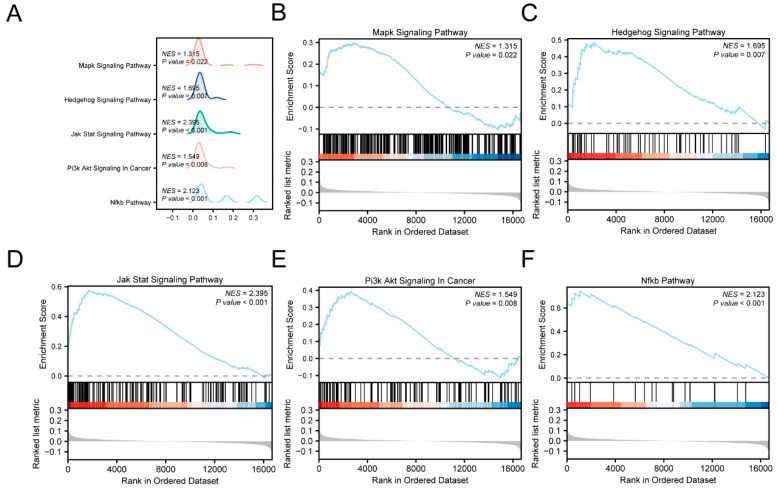
The GSEA for the combined datasets. (**A**) A ridge plot of the GSEA of the GSE114911 dataset. (**B**–**F**) The GSEA indicated a significant enrichment of the TRDEGs in several pathways: MAPK (**B**), Hedgehog (**C**), JAK-STAT (**D**), PI3k/Akt (**E**), and NF-κB (**F**). The screening criteria of the GSEA were a *p*-value < 0.05 and an FDR value (q-value) < 0.25.

**Figure 6 ijms-25-10100-f006:**
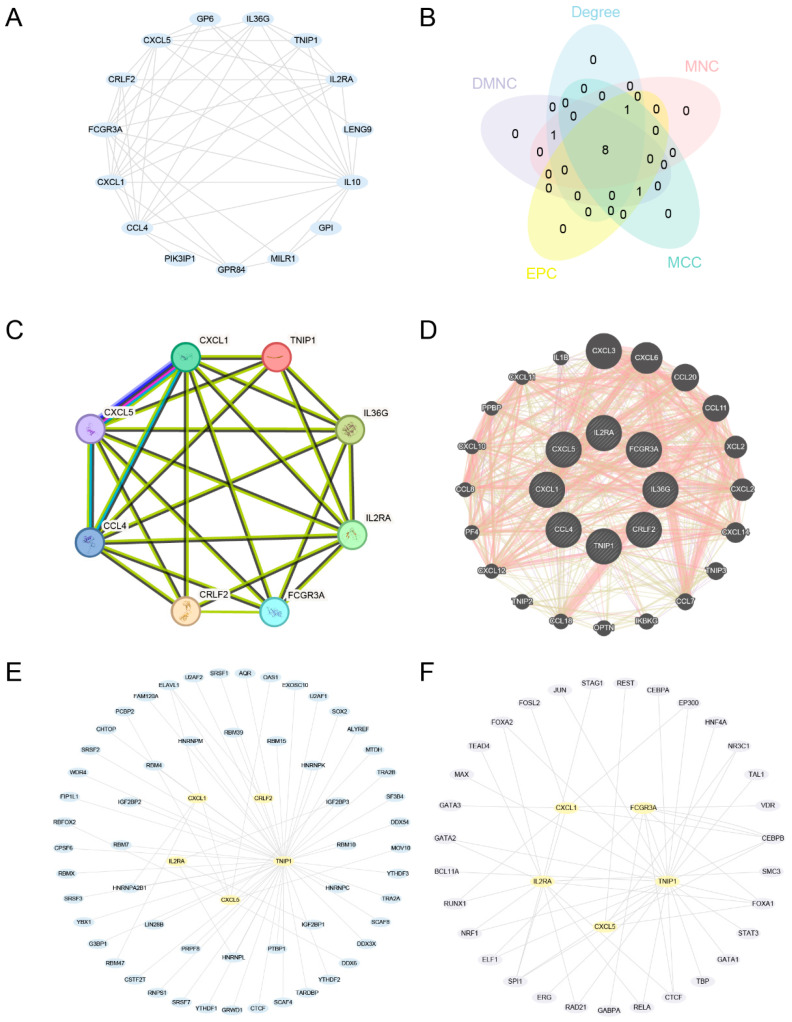
PPI, mRNA-RBP, and mRNA-TF interaction networks. (**A**) Network of TRDEGs based on PPIs. (**B**) Venn diagram illustrating the eight principal genes identified using MCC, MNC, EPC, Degree, and DMNC algorithms. (**C**) PPI network of the eight key genes. (**D**) Interaction network between the eight key genes and functionally similar genes. (**E**) Key genes–RBP interaction network. (**F**) Key genes–TF interaction network. Yellow ovals denote mRNA, blue ovals denote RBPs, and purple ovals denote TFs, respectively.

**Figure 7 ijms-25-10100-f007:**
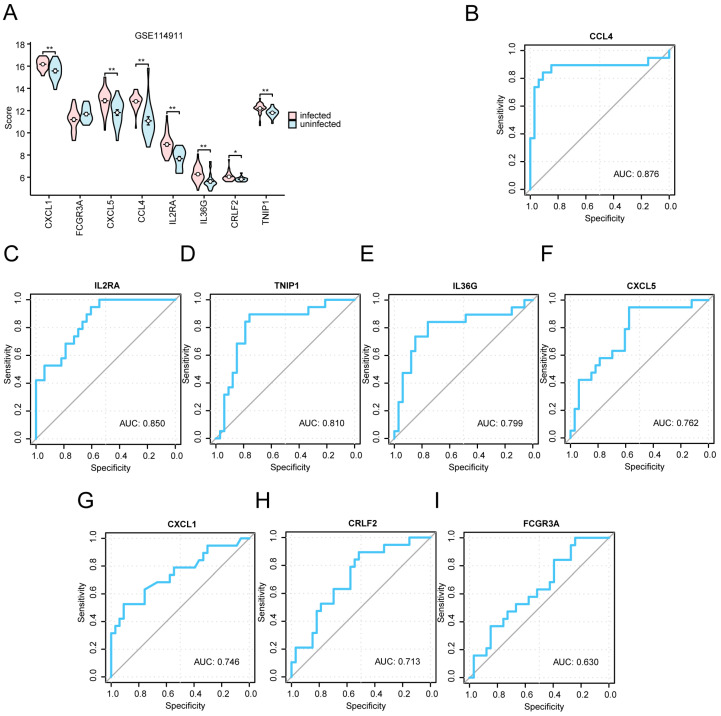
The expression difference analysis and ROC curves for key genes in the GSE114911 dataset between the infected and uninfected groups. (**A**) A comparative analysis of key genes in the datasets for infected versus uninfected groups. (**B**–**I**) ROC curve evaluations of the key genes *CCL4* (**B**), *IL2RA* (**C**), *TNIP1* (**D**), *IL36G* (**E**), *CXCL5* (**F**), *CXCL1* (**G**), *CRLF2* (**H**), and *FCGR3A* (**I**) in the GSE114911 dataset. ** denotes *p* < 0.01, indicating a high statistical significance; and * represents *p* < 0.05 and denotes statistical significance. The ROC curve in the AUC is close to 1, in order to better diagnose the results. The AUC indicated a certain accuracy when it fell in the range of 0.7–0.9. AUC values of 0.5~0.7 indicated a low accuracy. The infected group is represented in red, while the uninfected group is shown in blue.

**Figure 8 ijms-25-10100-f008:**
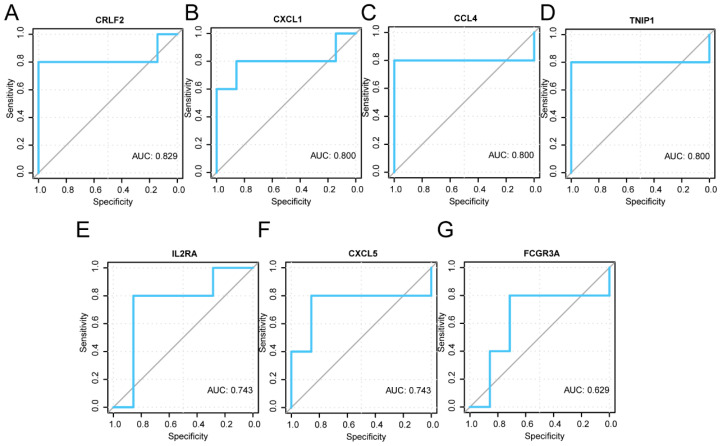
The ROC curves for key genes within the GSE139825 dataset between the infected and uninfected groups. (**A**–**G**) ROC curves for the key genes *CRLF2* (**A**), *CXCL1* (**B**), *CCL4* (**C**), *TNIP1* (**D**), *IL2RA* (**E**), *CXCL5* (**F**), and *FCGR3A* (**G**) in the GSE139825 dataset. AUC values of 0.7~0.9 demonstrated a certain accuracy. AUC values between 0.5 and 0.7 had a low accuracy.

**Figure 9 ijms-25-10100-f009:**
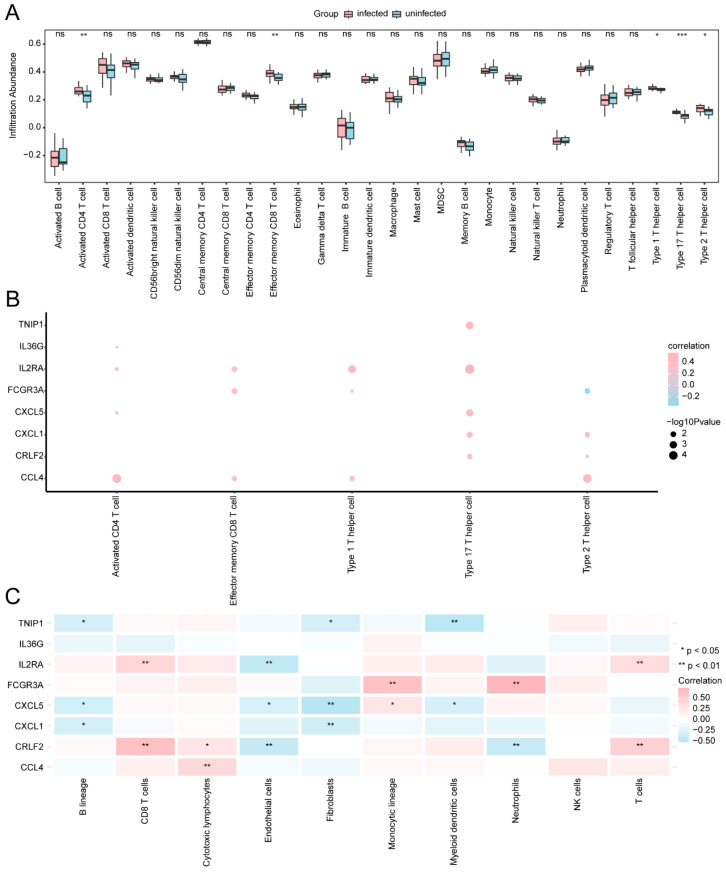
An analysis of immune infiltration (using an ssGSEA and MCPCounter). (**A**) A comparative diagram illustrating 28 immune cell types under the ssGSEA algorithm across different groups (infected/uninfected) within the GSE114911 dataset. (**B**) A heat map showing the correlation analysis results between key genes and the infiltration abundance of immune cells (*p* < 0.05), calculated by the ssGSEA algorithm. (**C**) The heat map display of the correlation analysis results between the key genes and the immune cell infiltration abundance was calculated by the MCPCounter algorithm. In the correlation heat map, the red circles represent a positive correlation between the genes and immune cell infiltration abundance, with a larger circle signifying a stronger correlation. The larger the circle, the stronger the correlation. The blue circles indicate a negative correlation between the genes and immune cell infiltration abundance, with larger circles reflecting a stronger correlation. ns denotes *p* > 0.05, indicating a lack of statistical significance. *** represents *p* < 0.001, signifying a very high level of statistical significance; ** denotes *p* < 0.01, indicating high statistical significance; and * represents *p* < 0.05 and denotes statistical significance.

**Table 1 ijms-25-10100-t001:** Characteristics of the study datasets.

GSE Series	Species	Platform	Total	Infected	Uninfected	PMID
GSE114911	*Homo sapiens*	GPL6480	52	33	19	29977236
GSE139825	*Homo sapiens*	GPL10558	12	7	5	32373118

## Data Availability

The datasets generated, coded, and analyzed during the current study are available upon request.

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
