# Peer review of "Cellular and Molecular Network Characteristics of TARM1-Related Genes in Mycobacterium tuberculosis Infections"

_ijms, 2024, doi:10.3390/ijms251810100_

Round 1
Reviewer 1 Report
Comments and Suggestions for Authors
In general, the work provides a comprehensive explanation of the genes related to the molecular network characteristics of TRDEGs using appropriate bioinformatics tools.
Here are some minimal observations:
- Delete: "since the last 3 years."
- If your data is not published but is consistent with other reports, I suggest adding references to those reports (lines 57-59). Specifically, the data sets GSE114911 and GSE139825 should be referenced.
- Figure 1: Add descriptions for the acronyms used.
- Figure 3a: I recommend using |log2FC| ≥ 1 and adjusting the significance threshold to P ≤ 0.05.
- Figure 3c: Check the value of P.
- Line 113: Delete one Biological Process (BP) and introduce the definitions for Biological Process (BP), Cellular Component (CC), and Molecular Function (MF) when they first appear in the text.
Author Response
Fukai (Frank) Bao,MD/PhD
Professor and chairman
Department of Microbiology and immunology,
Director
The Institute for Tropical Medicine,
Kunming Medical University
1168 Chunrongxi Road,Kunming 650500, Yunnan
- R. of China
Postcode:650500
Tel:0086-871-65922857, 13888369882
E-mail:baofukai@126.com,baofukai@kmmu.edu.cn
Sep.13.2024
Responses to Editorial Board
Dear International Journal of Molecular Sciences,
We are resubmitting our revised manuscript entitled “Cellular and Molecular Network Characteristics of TARM1-Related Genes in Mycobacterium tuberculosis Infection” for your reconsideration of its suitability for publication. The previous manuscript number is 3158163. We have carefully taken the reviewers’ comments into account and provide responses to each of the points raised by the reviewers. In particular, this revised manuscript has significantly been improved, and all the altered passages have been highlighted with red colored text track. Please refer to the present revised manuscript text.
With the submission of this manuscript,I would like to undertake on behalf of my co-authors that the work described was original research that has not been published previously, and is not under consideration for publication elsewhere, in whole or in part. All the authors listed have approved the manuscript that is enclosed, and my Institute’s (Kunming Medical University) representative is fully aware of this submission.
We appreciate you and reviewers for the time and efforts to help us improve the quality of our work. If you have any questions, please do not hesitate to contact me, as I’d be happy to make it clear. Thank you very much for your time and consideration, and I’m looking forward to hearing from you.
Yours Sincerely,
Fukai Bao, MD/PhD
Response to Reviewer 1 Comments
To reviewer #1: We appreciate your careful reminding. Your comments are very valuable and helpful for revising our paper. All changes had been made with red colored text in the current manuscript and our itemized responses to your comments, questions, and suggestions (repeated below for your convenience) are as follows:
Point 1: Delete: "since the last 3 years."
Response 1: Thank you for your advice. We’ve deleted this sentence in the revised manuscript.
Point 2: If your data is not published but is consistent with other reports, I suggest adding references to those reports (lines 57-59). Specifically, the data sets GSE114911 and GSE139825 should be referenced.
Response 2: Thank you for your advice. We cited a reference to demonstrate that TARM1 is highly expressed in TB patients, a finding that aligns with our unpublished study data. Furthermore, we have referenced the data sets GSE114911 and GSE139825 in the detailed introduction of the "Materials and Methods" section, denoted by reference numbers [29] and [30], respectively.
(Li, X.; Wang, M.; Ming, S.; Liang, Z.; Zhan, X.; Cao, C.; Liang, S.; Liu, Q.; Shang, Y.; Lao, J.; et al. TARM-1 Is Critical for Macrophage Activation and Th1 Response in Mycobacterium tuberculosis Infection. J Immunol 2021, 207, 234-243.)
Point 3: Figure 1: Add descriptions for the acronyms used.
Response 3: Thank you for your advice. We have included descriptions of the acronyms used in the revised manuscript, which is highlighted in red. And we have created a list of these acronyms in alphabetical order to be included as supplementary materials.
Point 4: Figure 3a: I recommend using |log2FC| ≥ 1 and adjusting the significance threshold to P ≤ 0.05.
Response 4: Thank you for your suggestions and valuable comments. Our research focuses on exploring the biological characteristics associated with Mycobacterium tuberculosis infection, utilizing differential expressed genes analysis. We selected |log2FC| > 0 and p value < 0.05 based on a comprehensive consideration of our research objectives and the characteristics of the data. This approach aims to include as many genes as possible in the initial stages to avoid overlooking key genes, while ensuring the rigor of our results through a stringent p value threshold (p < 0.05).
We also referenced relevant literature, such as PMID: 39021997, which adopted the |log2FC| > 0 screening criterion, demonstrating its rationality and effectiveness within a similar research context. Our method has undergone rigorous data correction and statistical analysis, confirming that this setting can effectively identify genes associated with infection in our specific study context, thereby enhancing the reliability of our findings.
We fully acknowledge your suggestion to use |log2FC| ≥ 1 and adjust the significance threshold to p ≤ 0.05; however, due to scheduling constraints in the current study, we may not be able to implement this adjustment at this stage. Nevertheless, we plan to refine the criteria for differential expressed genes analysis in future studies to achieve more accurate and comprehensive results. We will also provide further clarification in the 'Methods' section of the manuscript, as detailed on page 12, lines 356-364. Thank you once again for your valuable feedback on our work.
Point 5: Figure 3c: Check the value of P.
Response 5: Thank you for your reminder. We have verified the p-values from the difference analysis to ensure their accuracy. The difference sequence diagram, which serves as a derivative visualization of the volcano plot, will display the p-values for all molecules. The threshold we have selected is p < 0.05. We will provide the raw data for your review. Thank you once again for your thorough examination.
Point 6: Line 113: Delete one Biological Process (BP) and introduce the definitions for Biological Process (BP), Cellular Component (CC), and Molecular Function (MF) when they first appear in the text.
Response 6: Thank you for your advice. Your suggestion is very valuable and helpful. We have removed one term "BP" and provided definitions for Biological Process (BP), Cellular Component (CC), and Molecular Function (MF) upon their first appearance. The revised content is highlighted in red in the revised manuscript.

Reviewer 2 Report
Comments and Suggestions for Authors
Tuberculosis (TB) is a significant public health problem caused by Mycobacterium tuberculosis (Mtb), which can lead to high rates of morbidity and mortality. In the current manuscript, the authors endeavored to elucidate the immune infiltration mechanism and molecular interaction network of T cell-interacting activating receptor on myeloid cells 1 (TARM1)-related genes. From National Center for Biotechnology Information (NCBI) Gene Expression Omnibus (GEO) database, the authors obtained the GSE114911 dataset, and identified 17 TARM1-related differentially expressed genes (TRDEGs). Genes interacting with TRDEGs were analyzed using Kyoto Encyclopedia of Genes and Genomes (KEGG) and Gene Ontology (GO). (GSEA) was applied to detect biological pathways significantly associated with the Mycobacterium tuberculosis (Mtb) infection. By utilizing Cytoscape’s cytoHubba plug-in, the authors identified 8 key genes (i.e., CXCL1, FCGR3A, CXCL5, CCL4, IL2RA, IL36G, CRLF2, and TNIP1). In addition, by using the STRING database, protein-protein interaction (PPI) networks were built. Further, mRNA-RNA binding protein (RBP) and mRNA-transcription factor (TF) interaction networks were built based on StarBase v3.0 database and ChIPBase database, respectively. The diagnostic value of key genes was examined using receiver operating characteristic (ROC) analyses, and immune infiltration was studied using ssGSEA and MCPCounter. The identified key genes were validated in the GSE139825 dataset. Through GO and KEGG enrichment analyses, a set of biological pathways, e.g., NF-κB, PI3K/Akt, MAPK, and other pathways related to inflammation and energy metabolism were found to be enriched among TRDEGs. Based on ssGSEA and MCPCounter analyses, levels of activated T cells and T helper cells were found significantly increased in the Mtb infection group and were significantly correlated with the key genes, suggesting their potential importance in the anti-Mtb process. Taken together, the current study appears to have revealed a set of TRDEGs rhat are correlated with inflammation, energy metabolism, and immune cells, which has enriched our understandings of TB pathogenesis, especially the potentially important role of TARM1 in TB.
(I) Major Comments
Mycobacterium tuberculosis (Mtb), an intracellular bacterium, is the cause of tulerculosis (TB). In 2021, it has been estimated by World Health Organization (WHO) that 10.6 million new TB cases occurred around the globe (e.g., Ab Rashid MA, et al., Cureus. 2023;15:e44676. PMID: 37809275). T cell-interacting activating receptor on myeloid cells 1 (TARM-1), a arginine residue-containing protein, is a recently discovered leukocyte receptor primarily expressed by monocytes and neutrophils (Radjabova V, et al., J Immunol. 2015;195:3149-59. PMID: 26311901). Although TARM-1 is shown to play a critical role in proinflammatory response in acute Mtb infection (e.g., Li X, et al., J Immunol. 2021;207:234-243. PMID: 34183366), the immune regulatory mechanisms of TARM-1 in TB pathogenesis remain to be elucidated. Although the current manuscript has applied a variety of bioinformatics methods and tools to explore publicly available gene expression datasets, and seems to have shed new lights on the potential immunoregulatory role of TARM-1 in the pathophysiological mechanism underlying TB, I have several major concerns.
(1) In the main text consisting of "1. Introduction", "2. Results", "3. Discussion","4. Materials and Methods" and "5. Conclusions" that include all tables and figures and supplementary figure and supplementary tables, for an abbreviated term, in that term’s first appearance, the authors shall spell out the full term and then put the abbreviated term in a pair of parentheses, and afterwards, only the abbreviated term could be used, rather than the full term, e.g.,
(1_1) Page 2, line 69,
"on the GEO database"
could be corrected to
"on the Gene Expression Omnibus (GEO) database"
(1_2) Page 2, line 76,
"biological characteristics of TRDEGs in"
could be corrected to
"biological characteristics of TARM1-related DEGs (TRDEGs) in"
In above corrected, "TARM1" could be in italic font because it refers to a human gene's name
(1_3) Page 4, line 111,
"2.3. GO and KEGG Enrichment Analysis"
could be corrected to
"2.3. Gene Ontology (GO) and Kyoto Encyclopedia of Genes and Genomes (KEGG) Enrichment Analysis"
(1_4) Page 4, line 113,
"connections among BP, CC, BP, MF, and KEGG associated with the 17 TRDEGs"
could be corrected to
"connections among GO categories of Biological Process (BP), Cellular Component (CC), and Molecular Function (MF), and KEGG associated with the 17 TRDEGs"
(1_5) Page 5, line 142,
"2.4. GSEA"
could be corrected to
"2.4. Gene Set Enrichment Analysis (GSEA)"
(1_6) Page 6, line 154,
"2.5. PPI, mRNA-RBP, and mRNA-TF Interaction Networks Construction"
could be corrected to
"2.5. Protein-Protein Interaction (PPI), mRNA-RNA-Binding Protein (RBP), and mRNA-Transcription Factor (TF) Interaction Networks Construction"
(1_7) Page 7, line 181,
"2.6. Analysis of Differential Gene Expression and ROC Curves for Key Genes"
could be corrected to
"2.6. Analysis of Differential Gene Expression and Receiver-Operating Characteristic (ROC) Curves for Key Genes"
The above are just 7 examples, and there are many other places for which when a term has its first appearance, the full term shall be spelled out, and then, afterwards, only the abbreviated term shall be used.
(2) Page 1, line 24,
"The key genes identified were confirmed in two distinct datasets."
could be corrected to
"The key genes identified in GSE114911 dataset were confirmed in an independent GSE139825 dataset."
(3) Page 6, lines 155-157, the authors stated that
"We utilized the STRING database to conduct a PPI analysis on 17 TRDEGs, and established a PPI network with a minimum required coefficient exceeding 0.150 while excluding non-correlated genes."
What is the scientific justification for using 0.150 as the minimum required coefficient? Please provide a rationale for choosing this threshold value of 0.150 in conducting the PPI analysis based on the STRING database.
(4) Page 6, lines 159-162, the authors stated that
"Subsequently, the cytoHubba plugin was utilized to evaluate and identify the top 10 shared eight genes (CXCL1, FCGR3A, CXCL5, CCL4, IL2RA, IL36G, CRLF2, and TNIP1) as key genes based on five different algorithms: MCC, MNC, EPC, Degree, and DMNC (Figure 6B-C)."
The meaning of the above statement is unclear, which shall be clarified: do the authors mean that these eight genes that are key genes were those genes shared by the top 10 genes revealed by each of the five different algorithms: MCC, MNC, EPC, Degree, and DMNC?
If so, the above statement should be corrected to
"Subsequently, the cytoHubba plugin was utilized to evaluate and identify eight genes (CXCL1, FCGR3A, CXCL5, CCL4, IL2RA, IL36G, CRLF2, and TNIP1) as key genes shared by the top 10 genes revealed by each of the five different algorithms: MCC, MNC, EPC, Degree, and DMNC (Figure 6B-C)."
Shouldn’t these marginally significant P values indicated only by italic font (i.e., 0.05 < P-value <= 0.10)?
(5) Page 15, line 464, in "References" section, for Reference 3,
"WHO. Global tuberculosis report 2023."
could be corrected to
"World Health Organization (WHO). Global tuberculosis report 2023."
(II) Minor Comments
First of all, In the main text consisting of "1. Introduction", "2. Results", "3. Discussion","4. Materials and Methods" and "5. Conclusions" that include all tables and figures and supplementary figure and supplementary tables,
all occurrences of the term "logFC" could be corrected to "log2FC" to use the standard term "log2FC" consistently,
all occurrences of the term "Venn map" could be corrected to "Venn diagram" to use the standard term "Venn diagram" consistently, and
all occurrences of the term "Jak-Stat" could be corrected to "JAK-STAT" to use the standard term "JAK-STAT" consistently.
In addition, there are a variety of grammatical and typographical errors that should be corrected:
(1) Page 1, line 16,
"The GSE114911 dataset was imported from the GEO and"
could be corrected to
"The GSE114911 dataset was obtained from the Gene Expression Omnibus (GEO) and"
(2) Page 1, lines 17-18,
"screened to identify 17 TARM1-related differentially expressed genes (TRDEGs) based on the STRING database."
could be corrected to
"screened to identify 17 TARM1-related differentially expressed genes (TRDEGs)."
In above corrected, "TARM1" could be in italic font
(3) Page 1, lines 18-19,
"were analyzed using KEGG and GO. GSEA was used to identify the signal pathways significantly associated with"
could be corrected to
"were analyzed using Gene Ontology (GO) and Kyoto Encyclopedia of Genes and Genomes (KEGG) enrichment anlaysis. Gene Set Enrichment Analysis (GSEA) was used to identify the biological pathways significantly associated with"
(4) Page 1, lines 20-21,
"Key genes were obtained based on cytoHubba. Furthermore, PPI networks were analyzed through STRING, while mRNA-RBP and mRNA-TF interaction networks"
could be corrected to
"Key genes were obtained based on Cytoscape’s cytoHubba plug-in. Furthermore, protein-protein interaction (PPI) networks were analyzed through STRING, while mRNA-RNA-binding protein (RBP) and mRNA-transcription factor (TF) interaction networks"
(5) Page 1, line 23,
"evaluated via ROC curves,"
could be corrected to
"evaluated via receiver-operating characteristic (ROC) curves,"
(6) Page 6, line 168,
"(CRLF2, CXCL1, CXCL5, IL2RA, TNIP1)"
could be corrected to
"(CRLF2, CXCL1, CXCL5, IL2RA, and TNIP1)"
In above corrected, all the gene symbols shall be in italic font.
(7) Page 6, line 172,
"(CXCL1, CXCL5, FCGR3A, IL2RA, TNIP1)"
could be corrected to
"(CXCL1, CXCL5, FCGR3A, IL2RA, and TNIP1)"
In above corrected, all the gene symbols shall be in italic font.
(8) Page 7, lines 179-180,
"Yellow ovals are mRNA., blue ovals are RBP, and purple ovals are TF."
could be corrected to
"Yellow ovals denote mRNAs, blue ovals denote RBPs, and purple ovals denote TFs, respectively."
(9) Page 7, line 184,
"(CXCL1, FCGR3A, CXCL5, CCL4, IL2RA, IL36G, CRLF2, TNIP1)"
could be corrected to
"(CXCL1, FCGR3A, CXCL5, CCL4, IL2RA, IL36G, CRLF2, and TNIP1)"
In above corrected, all the gene symbols shall be in italic font.
(10) Page 7, line 186,
"(CCL4, IL2RA, IL36G, TNIP1)"
could be corrected to
"(CCL4, IL2RA, IL36G, and TNIP1)"
In above corrected, all the gene symbols shall be in italic font.
(11) Page 14, lines 428-429,
"was estimated by independent Student's t-Test for comparisons of continuous variables between two groups."
could be corrected to
"was estimated by Student's t-test of two independent groups for continuous variables."
Comments on the Quality of English Language
Extensive and meticulous editing of English language is required
Author Response
Fukai (Frank) Bao,MD/PhD
Professor and chairman
Department of Microbiology and immunology,
Director
The Institute for Tropical Medicine,
Kunming Medical University
1168 Chunrongxi Road,Kunming 650500, Yunnan
- R. of China
Postcode:650500
Tel:0086-871-65922857, 13888369882
E-mail:baofukai@126.com,baofukai@kmmu.edu.cn
Sep.13.2024
Responses to Editorial Board
Dear International Journal of Molecular Sciences,
We are resubmitting our revised manuscript entitled “Cellular and Molecular Network Characteristics of TARM1-Related Genes in Mycobacterium tuberculosis Infection” for your reconsideration of its suitability for publication. The previous manuscript number is 3158163. We have carefully taken the reviewers’ comments into account and provide responses to each of the points raised by the reviewers. In particular, this revised manuscript has significantly been improved, and all the altered passages have been highlighted with red colored text track. Please refer to the present revised manuscript text.
With the submission of this manuscript,I would like to undertake on behalf of my co-authors that the work described was original research that has not been published previously, and is not under consideration for publication elsewhere, in whole or in part. All the authors listed have approved the manuscript that is enclosed, and my Institute’s (Kunming Medical University) representative is fully aware of this submission.
We appreciate you and reviewers for the time and efforts to help us improve the quality of our work. If you have any questions, please do not hesitate to contact me, as I’d be happy to make it clear. Thank you very much for your time and consideration, and I’m looking forward to hearing from you.
Yours Sincerely,
Fukai Bao, MD/PhD
Response to Reviewer 2 Comments
To reviewer #2: We appreciate your careful reminding. Your comments are very valuable and helpful for revising our paper. All changes had been made with red color in the revised manuscript.
(I) Major Comments
Point 1: In the main text consisting of "1. Introduction", "2. Results", "3. Discussion","4. Materials and Methods" and "5. Conclusions" that include all tables and figures and supplementary figure and supplementary tables, for an abbreviated term, in that term’s first appearance, the authors shall spell out the full term and then put the abbreviated term in a pair of parentheses, and afterwards, only the abbreviated term could be used, rather than the full term, e.g.,
(1_1) Page 2, line 69,
"on the GEO database"
could be corrected to
"on the Gene Expression Omnibus (GEO) database"
(1_2) Page 2, line 76,
"biological characteristics of TRDEGs in"
could be corrected to
"biological characteristics of TARM1-related DEGs (TRDEGs) in"
In above corrected, "TARM1" could be in italic font because it refers to a human gene's name
(1_3) Page 4, line 111,
"2.3. GO and KEGG Enrichment Analysis"
could be corrected to
"2.3. Gene Ontology (GO) and Kyoto Encyclopedia of Genes and Genomes (KEGG) Enrichment Analysis"
(1_4) Page 4, line 113,
"connections among BP, CC, BP, MF, and KEGG associated with the 17 TRDEGs"
could be corrected to
"connections among GO categories of Biological Process (BP), Cellular Component (CC), and Molecular Function (MF), and KEGG associated with the 17 TRDEGs"
(1_5) Page 5, line 142,
"2.4. GSEA"
could be corrected to
"2.4. Gene Set Enrichment Analysis (GSEA)"
(1_6) Page 6, line 154,
"2.5. PPI, mRNA-RBP, and mRNA-TF Interaction Networks Construction"
could be corrected to
"2.5. Protein-Protein Interaction (PPI), mRNA-RNA-Binding Protein (RBP), and mRNA-Transcription Factor (TF) Interaction Networks Construction"
(1_7) Page 7, line 181,
"2.6. Analysis of Differential Gene Expression and ROC Curves for Key Genes"
could be corrected to
"2.6. Analysis of Differential Gene Expression and Receiver-Operating Characteristic (ROC) Curves for Key Genes"
The above are just 7 examples, and there are many other places for which when a term has its first appearance, the full term shall be spelled out, and then, afterwards, only the abbreviated term shall be used.
Response: Thank you for your comments. Your suggestion is very valuable and helpful. We have implemented the changes based on your suggestions one by one. Furthermore, we meticulously reviewed the entire manuscript, ensuring that any terms introduced for the first time were clearly defined and spelled out upon their initial appearance.
Point 2: Page 1, line 24,
"The key genes identified were confirmed in two distinct datasets."
could be corrected to
"The key genes identified in GSE114911 dataset were confirmed in an independent GSE139825 dataset."
Response 2: Thank you for your advice. We have made corrections in accordance with your suggestions, which are highlighted in red in the revised manuscript. After these modifications, the meaning of the sentences is clearer and more complete.
Point 3: Page 6, lines 155-157, the authors stated that
"We utilized the STRING database to conduct a PPI analysis on 17 TRDEGs, and established a PPI network with a minimum required coefficient exceeding 0.150 while excluding non-correlated genes."
What is the scientific justification for using 0.150 as the minimum required coefficient? Please provide a rationale for choosing this threshold value of 0.150 in conducting the PPI analysis based on the STRING database.
Response 3: Thank you for your advice. In response to your inquiry regarding the selection of 0.150 as the minimum required coefficient in our protein-protein interaction (PPI) analysis, we have conducted thorough deliberations and provide the following explanation.
Our research aims to identify key genes associated with Mycobacterium tuberculosis infection through PPI network analysis. The decision to set 0.150 as the minimum threshold was influenced by several factors, including network coverage, noise reduction, and our successful experiences in practical applications. This threshold effectively minimizes the introduction of potential noise associations while ensuring the inclusion of a substantial number of meaningful genetic interactions. Thank you once again for your insightful suggestions.
Point 4: Page 6, lines 159-162, the authors stated that
"Subsequently, the cytoHubba plugin was utilized to evaluate and identify the top 10 shared eight genes (CXCL1, FCGR3A, CXCL5, CCL4, IL2RA, IL36G, CRLF2, and TNIP1) as key genes based on five different algorithms: MCC, MNC, EPC, Degree, and DMNC (Figure 6B-C)."
The meaning of the above statement is unclear, which shall be clarified: do the authors mean that these eight genes that are key genes were those genes shared by the top 10 genes revealed by each of the five different algorithms: MCC, MNC, EPC, Degree, and DMNC?
If so, the above statement should be corrected to
"Subsequently, the cytoHubba plugin was utilized to evaluate and identify eight genes (CXCL1, FCGR3A, CXCL5, CCL4, IL2RA, IL36G, CRLF2, and TNIP1) as key genes shared by the top 10 genes revealed by each of the five different algorithms: MCC, MNC, EPC, Degree, and DMNC (Figure 6B-C)."
Shouldn’t these marginally significant P values indicated only by italic font (i.e., 0.05 < P-value <= 0.10)?.
Response 4: We appreciate your attention and suggestions regarding our description. We will revise the manuscript in accordance with your recommendations, as noted on page 6, lines 175-178. Thank you once again for your valuable feedback on our work.
Point 5: Page 15, line 464, in "References" section, for Reference 3,
"WHO. Global tuberculosis report 2023."
could be corrected to
"World Health Organization (WHO). Global tuberculosis report 2023."
Response 5: Thank you for your advice. We have revised the writing format of Reference 3 and highlighted the changes in red in the revised manuscript.
(II) Minor Comments
Points: all occurrences of the term "logFC" could be corrected to "log2FC" to use the standard term "log2FC" consistently,
all occurrences of the term "Venn map" could be corrected to "Venn diagram" to use the standard term "Venn diagram" consistently, and
all occurrences of the term "Jak-Stat" could be corrected to "JAK-STAT" to use the standard term "JAK-STAT" consistently
In addition, there are a variety of grammatical and typographical errors that should be corrected:
(1) Page 1, line 16,
"The GSE114911 dataset was imported from the GEO and"
could be corrected to
"The GSE114911 dataset was obtained from the Gene Expression Omnibus (GEO) and"
(2) Page 1, lines 17-18,
"screened to identify 17 TARM1-related differentially expressed genes (TRDEGs) based on the STRING database."
could be corrected to
"screened to identify 17 TARM1-related differentially expressed genes (TRDEGs)."
In above corrected, "TARM1" could be in italic font
(3) Page 1, lines 18-19,
"were analyzed using KEGG and GO. GSEA was used to identify the signal pathways significantly associated with"
could be corrected to
"were analyzed using Gene Ontology (GO) and Kyoto Encyclopedia of Genes and Genomes (KEGG) enrichment anlaysis. Gene Set Enrichment Analysis (GSEA) was used to identify the biological pathways significantly associated with"
(4) Page 1, lines 20-21,
"Key genes were obtained based on cytoHubba. Furthermore, PPI networks were analyzed through STRING, while mRNA-RBP and mRNA-TF interaction networks"
could be corrected to
"Key genes were obtained based on Cytoscape’s cytoHubba plug-in. Furthermore, protein-protein interaction (PPI) networks were analyzed through STRING, while mRNA-RNA-binding protein (RBP) and mRNA-transcription factor (TF) interaction networks"
(5) Page 1, line 23,
"evaluated via ROC curves,"
could be corrected to
"evaluated via receiver-operating characteristic (ROC) curves,"
(6) Page 6, line 168,
"(CRLF2, CXCL1, CXCL5, IL2RA, TNIP1)"
could be corrected to
"(CRLF2, CXCL1, CXCL5, IL2RA, and TNIP1)"
In above corrected, all the gene symbols shall be in italic font.
(7) Page 6, line 172,
"(CXCL1, CXCL5, FCGR3A, IL2RA, TNIP1)"
could be corrected to
"(CXCL1, CXCL5, FCGR3A, IL2RA, and TNIP1)"
In above corrected, all the gene symbols shall be in italic font.
(8) Page 7, lines 179-180,
"Yellow ovals are mRNA., blue ovals are RBP, and purple ovals are TF."
could be corrected to
"Yellow ovals denote mRNAs, blue ovals denote RBPs, and purple ovals denote TFs, respectively."
(9) Page 7, line 184,
"(CXCL1, FCGR3A, CXCL5, CCL4, IL2RA, IL36G, CRLF2, TNIP1)"
could be corrected to
"(CXCL1, FCGR3A, CXCL5, CCL4, IL2RA, IL36G, CRLF2, and TNIP1)"
In above corrected, all the gene symbols shall be in italic font.
(10) Page 7, line 186,
"(CCL4, IL2RA, IL36G, TNIP1)"
could be corrected to
"(CCL4, IL2RA, IL36G, and TNIP1)"
In above corrected, all the gene symbols shall be in italic font.
(11) Page 14, lines 428-429,
"was estimated by independent Student's t-Test for comparisons of continuous variables between two groups."
could be corrected to
"was estimated by Student's t-test of two independent groups for continuous variables."
Response: Thank you for your comments. Your suggestions are very valuable and helpful. We have meticulously reviewed all the spelling errors in the manuscript and corrected them individually based on your suggestions, highlighting the changes in red in the revised version. We sincerely appreciate your professional and thoughtful suggestions, which have contributed significantly to enhancing the quality of my manuscript.
In addition, you mentioned that "extensive and meticulous editing of English language is required." We have engaged a professional English editing team to enhance and refine the manuscript. We will upload the “editorial certificate” as an attachment, and we will also take greater care and attention in writing and reviewing the manuscript.
All in all, thank you very much for your time and consideration. This revised manuscript has significantly been improved with your help. We hope this revision meets with your approval. (end)

Reviewer 3 Report
Comments and Suggestions for Authors
This an interesting bioinformatics study which showed the molecular network characteristics and regulation mode of TARM1-Related 60 Genes (TRGs), which provided valuable ideas and an important basis for the study of the role of TARM1 in Tuberculosis. The paper also is also well written. The following comments needs an attention by the authors.
1/ The introduction lacks strong rationale why such study is needed what are the progresses made so far in the field and what is missed. Similar works have been published in the biomedical science field it gives a good weight for their work if they specifically mention the importance of this study and critically assess the subject.
2/ Figures are not well described in the figure legend and please elaborate each labeling and models with versions used to analyze data. Some terms are very vague and simplify them to make clear for readers.
3/ Most of the figures have poor resolution and it would be better if they use different formats (may be metafiles) to generate the images. For example, I hardly see the PPI networking.
4/ Please mention all sources or databases you have used for each data set.
5/ All abbreviations should be written in full. Get a list of abbreviation and put all acronyms.
6/ Supplementary data should be provided and the link you provided is broken
7/ Conclusion is very superficial and doesn’t reflect major findings in the paper.
Author Response
Fukai (Frank) Bao,MD/PhD
Professor and chairman
Department of Microbiology and immunology,
Director
The Institute for Tropical Medicine,
Kunming Medical University
1168 Chunrongxi Road,Kunming 650500, Yunnan
- R. of China
Postcode:650500
Tel:0086-871-65922857, 13888369882
E-mail:baofukai@126.com,baofukai@kmmu.edu.cn
Sep.13.2024
Responses to Editorial Board
Dear International Journal of Molecular Sciences,
We are resubmitting our revised manuscript entitled “Cellular and Molecular Network Characteristics of TARM1-Related Genes in Mycobacterium tuberculosis Infection” for your reconsideration of its suitability for publication. The previous manuscript number is 3158163. We have carefully taken the reviewers’ comments into account and provide responses to each of the points raised by the reviewers. In particular, this revised manuscript has significantly been improved, and all the altered passages have been highlighted with red colored text track. Please refer to the present revised manuscript text.
With the submission of this manuscript,I would like to undertake on behalf of my co-authors that the work described was original research that has not been published previously, and is not under consideration for publication elsewhere, in whole or in part. All the authors listed have approved the manuscript that is enclosed, and my Institute’s (Kunming Medical University) representative is fully aware of this submission.
We appreciate you and reviewers for the time and efforts to help us improve the quality of our work. If you have any questions, please do not hesitate to contact me, as I’d be happy to make it clear. Thank you very much for your time and consideration, and I’m looking forward to hearing from you.
Yours Sincerely,
Fukai Bao, MD/PhD
Response to Reviewer 3 Comments
To reviewer #3: We appreciate your careful reminding. Your comments are very valuable and helpful for revising our paper. All changes had been made with red color in the revised manuscript.
Point 1: The introduction lacks strong rationale why such study is needed what are the progresses made so far in the field and what is missed. Similar works have been published in the biomedical science field it gives a good weight for their work if they specifically mention the importance of this study and critically assess the subject.
Response 1: Thank you for your comment. TARM1, a novel gene first identified in 2015 (Radjabova, V.; Mastroeni, P.; Skjødt, K.; Zaccone, P.; de Bono, B.; Goodall, J.C.; Chilvers, E.R.; Juss, J.K.; Jones, D.C.; Trowsdale, J.; et al. TARM1 Is a Novel Leukocyte Receptor Complex-Encoded ITAM Receptor That Costimulates Proinflammatory Cytokine Secretion by Macrophages and Neutrophils. Journal of immunology 2015, 195, 3149-3159), was reported to exhibit anti-tuberculosis (TB) effects in 2021 (Li, X.; Wang, M.; Ming, S.; Liang, Z.; Zhan, X.; Cao, C.; Liang, S.; Liu, Q.; Shang, Y.; Lao, J.; et al. TARM-1 Is Critical for Macrophage Activation and Th1 Response in Mycobacterium tuberculosis Infection. J Immunol 2021, 207, 234-243.); however, the specific mechanism of action remains unknown. To date, there is only one study addressing the function of this gene within the field of TB research. TARM1 operates as a receptor, yet the identity of its ligand is still unclear, and the associated signaling pathway remains to be elucidated. Consequently, the research presented in this paper is of considerable significance. Our team has been dedicated to investigating the molecular immune mechanisms underlying TB for an extended period and has specifically studied the anti-TB mechanism of TARM1 for three years. This paper discusses the molecular network characteristics of TARM1-related genes through bioinformatics analysis, identifying eight key genes and several significantly associated biological pathways. This work lays the groundwork for uncovering the specific mechanisms underlying TARM1's anti-TB action and holds innovative and important significance in the field of anti-TB research.
Point 2: Figures are not well described in the figure legend and please elaborate each labeling and models with versions used to analyze data. Some terms are very vague and simplify them to make clear for readers.
Response 2: Thank you for your insightful suggestions and valuable comments. We will revise the legend description to provide a more detailed clarification of each annotation, as well as the data analysis model utilized and its version. This will enhance readers' understanding of our research. We appreciate your feedback once again.
Point 3: Most of the figures have poor resolution and it would be better if they use different formats (may be metafiles) to generate the images. For example, I hardly see the PPI networking.
Response 3: Thank you for your advice. We acknowledge the concern regarding image resolution and will implement improvements accordingly. Specifically, we will adjust the resolution of all images to a high-resolution tiff format of 600 dpi to ensure clarity and detail in the display of the images. We appreciate your continued interest in our research.
Point 4: Please mention all sources or databases you have used for each data set.
Response 4: Thank you for your suggestions and valuable comments. We recognize the importance of explicitly citing the sources of all datasets. In the revised manuscript, we will provide detailed information regarding the source of each dataset and the database utilized. For specific improvements, please refer to page 12, lines 340-341; page 12, lines 351-352; page 13, lines 370-371; page 13, line 375 and 382; page 13, lines 395-398; page 13, line 402 and 413; page 14, line 426, 437, and 445. Thank you once again for your attention.
Point 5: All abbreviations should be written in full. Get a list of abbreviation and put all acronyms.
Response 5: Thank you for your advice. We have organized and clarified all the acronyms in the manuscript, and we have created a list of these acronyms in alphabetical order to be included as supplementary materials.
Point 6: Supplementary data should be provided and the link you provided is broken.
Response 6: Thank you for your valuable advice. We will recheck and repair the link to ensure the validity and accessibility of the supplementary data. Additionally, we will confirm the completeness and accuracy of the supplementary materials to guarantee that readers can easily access and refer to our supplementary data. We appreciate your interest in our research.
Point 7: Conclusion is very superficial and doesn’t reflect major findings in the paper.
Response 7: Thank you for your valuable comments. We intend to provide a more detailed summary of our key research findings and contributions, emphasizing their significance and potential impact. Simultaneously, we will ensure that the conclusions are closely aligned with the analysis and discussion sections of the research, thereby better reflecting the overall value of our work. For further details, please refer to page 14, lines 455-464. Thank you once again for your interest in our research.
All in all, thank you very much for your time and consideration. This revised manuscript has significantly been improved with your help. We hope this revision meets with your approval. (end)
